# Cumulative incidence of midline incisional hernia and its surgical treatment after radical cystectomy and urinary diversion for bladder cancer: A nation-wide population-based study

Fredrik Liedberg[1,2]*, Oskar Hagberg[2,3], Firas Aljabery[4], Truls Gårdmark[5], Staffan Jahnson[4], Tomas Jerlström[6], Agneta Montgomery[7], Amir Sherif[8], Viveka Ströck[9], Christel Häggström[10,11], Lars Holmberg[11,12]

1 Department of Urology Skåne University Hospital, Malmö, Sweden, 2 Institution of Translational Medicine, Lund University, Malmö, Sweden, 3 Regional Cancer Centre South, Region Skåne, Lund, Sweden, 4 Division of Urology, Department of Clinical and Experimental Medicine, Linköping University, Linköping, Sweden, 5 Department of Clinical Sciences, Danderyd Hospital, Karolinska Institute, Stockholm, Sweden, 6 Department of Urology, School of Medical Sciences, Faculty of Medicine and Health, Örebro University, Örebro, Sweden, 7 Institution of Clinical Sciences Malmö, Surgical Research Unit, Lund University, Lund, Sweden, 8 Department of Surgical and Perioperative Sciences, Urology and Andrology, Umeå University, Umeå, Sweden, 9 Department of Urology, Sahlgrenska University Hospital and Institute of Clinical Sciences, Sahlgrenska Academy, University of Gothenburg, Gothenburg, Sweden, 10 Department of Biobank Research, Umeå University, Umeå, Sweden, 11 Department of Surgical Sciences, Uppsala University, Uppsala, Sweden, 12 School of Cancer and Pharmaceutical Sciences, King's College London, London, United Kingdom

* fredrik.liedberg@med.lu.se

## Abstract

### Background and objective

To study the cumulative incidence and surgical treatment of midline incisional hernia (MIH) after cystectomy for bladder cancer.

### Methods

In the nationwide Bladder Cancer Data Base Sweden (BladderBaSe), cystectomy was performed in 5646 individuals. Cumulative incidence MIH and surgery for MIH were investigated in relation to age, gender, comorbidity, previous laparotomy and/or inguinal hernia repair, operative technique, primary/secondary cystectomy, postoperative wound dehiscence, year of surgery, and period-specific mean annual hospital cystectomy volume (PSMAV).

### Results

Three years after cystectomy the cumulative incidence of MIH and surgery for MIH was 8% and 4%, respectively. The cumulative incidence MIH was 12%, 9% and 7% in patients having urinary diversion with continent cutaneous pouch, orthotopic neobladder and ileal conduit. Patients with postoperative wound dehiscence had a higher three-year cumulative incidence MIH (20%) compared to 8% without. The corresponding cumulative incidence

**Data Availability Statement:** Data cannot be shared publicly because of patient-related and are confidential. The data in in BladderBaSe is partly available in annual reports from the Swedish National Registry of Urinary Bladder Cancer (SNRUBC) and also are accessible online at https://statistik.incanet.se/urinblasecancer/. Collaborators can propose and apply for studies in the BladderBaSe using a standardised form. After approved application, the project data administrators can upload study-specific files with selected variables to a server for statistical analysis through remote access. Users of this system will be charged for software licences, data administration and for preprocessing of study files. For more information contact either the PI BladderBaSe (lars.holmberg@kcl.ac.uk) or the corresponding author (fredrik.liedberg@med.lu.se).

**Funding:** This work was supported by the Swedish Cancer Society (grant numbers CAN 2019/62 and CAN 2017/278), Lund Medical Faculty (ALF), Skåne University Hospital Research Funds, the Gyllenstierna Krapperup's Foundation, Skåne County Council's Research and Development Foundation (REGSKANE-622351), Gösta Jönsson Research Foundation, The Foundation of Urological Research (Ove and Carin Carlsson bladder cancer donation) and Hillevi Fries Research Foundation. The funding sources had no role in the study design, data analyses, interpretation or writing the manuscript.

**Competing interests:** NO authors have competing interests.

surgery for MIH three years after cystectomy was 9%, 6%, and 4% for continent cutaneous, neobladder, and conduit diversion, respectively, and 11% for individuals with postoperative wound dehiscence (vs 4% without). Using multivariable Cox regression, secondary cystectomy (HR 1.3 (1.0–1.7)), continent cutaneous diversion (HR 1.9 (1.1–2.4)), robot-assisted cystectomy (HR 1.8 (1–3.2)), wound dehiscence (HR 3.0 (2.0–4.7)), cystectomy in hospitals with PSMAV 10–25 (HR 1.4 (1.0–1.9)), as well as cystectomy during later years (HRs 2.5–3.1) were all independently associated with increased risk of MIH.

## Conclusions

The cumulative incidence of MIH was 8% three years postoperatively, and increase over time. Avoiding postoperative wound dehiscence after midline closure is important to decrease the risk of MIH.

## Introduction

The occurrence of midline incisional hernia (MIH) after radical cystectomy and urinary diversion for bladder cancer or the proportion of these patients that require surgical repair are scarcely reported in the literature [1]. Published studies have often been hampered by either reporting simultaneously on different long-term complications after cystectomy or by a single-center design [2, 3]. Furthermore, the number of patients included in previous reports is limited. Thus far, only one population-based study has studied MIH after cystectomy [4]. This study considered only MIH complications requiring in-hospital care (but not necessarily surgery) as endpoint [4]. A total of one or several events were reported in 2.6% of the patients. In one United States tertial referral centre the risk of having a MIH diagnosis at end of follow-up was 19% [1].

Our objective was to conduct a population-based nationwide study including all bladder cancer patients subjected to radical cystectomy reported to BladderBaSe 1997 through 2014 to estimate the incidence of MIH development and the proportion of such hernias that led to surgical treatment. Furthermore, we aimed to identify risk factors for developing a MIH and having surgery for MIH.

## Material and methods

### Study design and participants

The principles of data extraction applied and linkages of the current study are described in detail in the BladderBaSe cohort profile [5]. All 5646 individuals diagnosed with bladder cancer in Sweden during the period 1997–2014 and subsequently treated with radical cystectomy, either at diagnosis as primary treatment (n = 4108) or later at progression as secondary treatment (n = 1538) were included.

Patients with ICD-10 codes for MIH (K43.9/K43.0/K43.1/K43.0A/K.43.0B/K43.1A/K43.1B/K.43.2/K43.2A/K.43.2) were identified from the hospital inpatient and outpatient registries, thus without knowing if a clinical examination and/or a radiological investigation was used to diagnose MIHs. Patients who had surgery for MIH after the date of cystectomy was ascertained from the inpatient registry by using the following ICD-10 codes: JAD10/JAD11/JAD13/JAD20/JAD23/JAD30/JAD33/JAD40/JAD41/JAD43/JAD47/JAD50/JAD51/JAD60/JAD61/JAD63/JAD67/JAD70/JAD71/JAD73/JAD80/JAD81/JAD84/JAD87. However, no

information on the indications that were applied for performing surgery for MIH or surgical techniques used were available.

We investigated if the diagnosis and/or surgery for a MIH were associated with age, gender, comorbidity, previous laparotomy, previous inguinal or incisional hernia repair before cystectomy, cystectomy as primary treatment at diagnosis or as secondary treatment at disease progression, type of urinary diversion (continent cutaneous pouch, orthotopic neobladder, or ileal conduit) surgical technique (open or robotic-assisted cystectomy), postoperative occurrence of wound dehiscence (diagnosis and/or surgery) within 60 days of surgery (a timepoint after which wound dehiscence not likely is related to the cystectomy) and hospital cystectomy volume. All these variables were ascertained either from the SNRUBC or from the inpatient or outpatient registries. Comorbidity was measured using the Charlson Comorbidity Index (CCI), and was calculated based on a list of diseases using a specific weight assigned to each disease category according to data from the national inpatient and outpatient register [5]. Information about whether early recovery after surgery (ERAS) measures were applied before, during and after radical cystectomy was lacking.

To investigate the association between hospital volume and MIH diagnosis and/or surgery, the Period-Specific Mean Annual Volume (PSMAV) was calculated [6]. PSMAV was defined as the cystectomy volume per year for 3 years preceding radical cystectomy based on all patients subjected to such surgery during the study period. PSMAV was stratified in tertiles adjusted to the prevailing literature on volume-outcome, i.e.: [0–10), [10–25) and [25–86] annual cystectomies [6].

The individual patient's date for radical cystectomy was used as the starting point for follow-up. Date of death, emigration, or 31 December 2014, was regarded as the end of follow-up, whatever happened first.

All data used were fully anonymized during the study that was approved by the Research Ethics Board of Uppsala University, Sweden (File no. 2015/277).

### Statistical analysis

The cumulative incidence of MIH and surgery for MIH was calculated using standard Kaplan-Meier technique, with date of cystectomy as starting point and censoring for loss to follow-up or death. The chi square test or Fischer's exact test was used to compare proportions MIH for categorical variables. The t-test was applied to compare continuous variables between groups. Univariate and multivariate Cox regression models were used to investigate the associations between preoperative risk factors for MIH and surgery during follow-up. Patients were censored at lost to follow-up and death, just as when the cumulative incidence was computed. Analyses were performed with the R statistical package version 3.4.2. ((R Core Team ☉2017]). (R: A language and environment for statistical computing. R Foundation for Statistical Computing, Vienna, Austria URL https://www.r-project.org).

### Results

Median age at cystectomy and urinary tract reconstruction among the 5646 individuals operated in Sweden between 1997 and 2014 was 69 (inter quartile range (IQR) 63–75) years. A total of 1350 (24%) were females. Median follow-up time was 2.3 (IQR 0.9–5.8) years. Patient characteristics including number of patients with potential pre- and/or perioperative risk factors for midline incisional hernia (MIH) formation are reported in Table 1.

A majority of patients were operated with open cystectomy (94%) and 346 (6%) were subjected to robotic-assisted radical cystectomy. The proportion of patients receiving robotic-

**Table 1. Number of patients with and without midline incisional hernia diagnosis and surgery.**

| Patient characteristics | Numbers with midline incisional hernia diagnosis | Numbers with midline incisional hernia surgery | Numbers without midline incisional hernia diagnosis or surgery | Total numbers |
|---|---|---|---|---|
| | (n = 379) | (n = 205) | (n = 5267) | (n = 5646) |
| Gender: | | | | |
| Male | 281 | 157 | 4010 | 4291 (76%) |
| Female | 98 | 48 | 1257 | 1355 (24%) |
| Median age at cystectomy (Interquartile Range) years: | 67 (62–73) | 66 (60–71) | 70 (63–75)* | 69 (63–75) |
| CCI: | | | | |
| 0 | 275 | 158 | 3731 | 4006 (71%) |
| 1 | 54 | 24 | 707 | 761 (13%) |
| 2 | 35 | 16 | 521 | 556 (10%) |
| 3 | 6 | 5 | 139 | 145 (3%) |
| >3 | 4 | 1 | 113 | 117 (2%) |
| missing | 5 | 1 | 56 | 61 (1%) |
| Previous laparotomy: | | | | |
| No | 377 | 204 | 5213 | 5590 (99%) |
| Yes | 2 | 1 | 54 | 56 (1%) |
| Previous inguinal hernia repair: | | | | |
| No | 373 | 202 | 5142 | 5515 (98%) |
| Yes | 6 | 3 | 125 | 131 (2%) |
| Incisional hernia before cystectomy: | | | | |
| No | 376 | 204 | 5237 | 5613 (99%) |
| Yes | 3 | 1 | 30 | 33 (1%) |
| Primary cystectomy | 251 | 132 | 3857 | 4108 (73%) |
| Secondary cystectomy | 128 | 73 | 1410 | 1538 (27%) |
| Urinary diversion: | | | | |
| Ileal conduit | 250 | 111 | 4097 | 4347 (77%) |
| Orthtopic neobladder | 72 | 53 | 798 | 870 (15%) |
| Continent cutaneous pouch | 57 | 41 | 372 | 429 (8%) |
| Robot-assisted cystectomy: | | | | |
| No | 362 | 194 | 4938 | 5300 (94%) |
| Yes | 17 | 11 | 329 | 346 (6%) |
| Wound dehiscence within 60 days after cystectomy: | | | | |
| No | 353 | 189 | 5112 | 5465 (97%) |
| Yes | 26 | 16 | 155 | 181 (3%) |
| PSMAV: | | | | |
| [0–10), | 87 | 46 | 1497 | 1584 (28%) |
| [10–25), | 170 | 96 | 2036 | 2206 (39%) |
| [25–86], | 117 | 60 | 1691 | 1808 (32%) |
| missing | 5 | 3 | 43 | 48 (1%) |

PSMAV = Period-specific mean annual hospital volume, CCI = Charlson comorbidity index. (*p<0.001 (t-test)).

assisted surgery increased during the last ten years in the study from 0% to 29% (122/425) during 2014.

A total of 379 patients were diagnosed with a MIH after cystectomy and surgery was performed in 205 individuals. The cumulative incidence of patients diagnosed with a MIH a

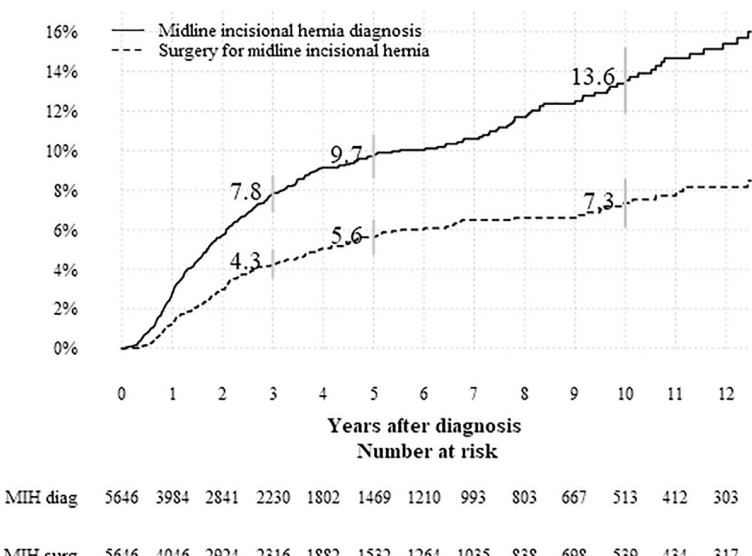

**Fig 1. Cumulative incidence with 95% confidence interval at selected time points.** The cumulative proportion of patients diagnosed with and operated for midline incisional hernia, respectively. Grey bars represent 95% CI at three, five and ten years.

three, five and ten years was 8%, 10%, and 14%, respectively, and the corresponding cumulative incidences of having surgery for MIH was 4%, 6%, and 7% (Fig 1).

Fig 2 shows the cumulative incidence of MIH diagnosed and repaired at three years, as stratified by three-year time-periods. Patients with cystectomy during the last three-year strata 2012–2014 were excluded in Fig 2, as they did not have three years follow-up time. The cumulative incidence MIH at three years follow-up increased during the study period, however the cumulative incidence of patients subjected to surgery for MIH at three years follow-up decreased.

Continent urinary diversion with continent cutaneous pouch or orthotopic neobladder were at three years postoperatively associated with larger proportions of MIH, 12% and 9% respectively, compared to urinary diversion with an ileal conduit (7% (p<0.03)). The corresponding proportions for open and robotic assisted cystectomy were 8% and 14%, respectively (p = 0.06). A larger proportion of patients suffering from wound dehiscence after cystectomy were diagnosed with MIH at three years after cystectomy (20%) compared to 8% without postoperative wound dehiscence (p<0.001). For patients operated in hospitals with a period-specific mean annual volume (PSMAV) in the upper two tertiles (10–25 and 25–86 annual cystectomies, respectively), a higher proportion of patients were diagnosed with MIH at three years (9%) compared to 6% in the lower PSMAV tertile (p = 0.04). Other potential risk-factors for MIH development are displayed in Table 1, however the cumulative incidence of MIH diagnosis were similar between groups.

Patients subjected to surgery for MIH were younger at cystectomy than those not being diagnosed or operated for midline incisional hernia (Table 1). Surgery for MIH was more frequently performed in patients receiving continent cutaneous diversion (9%) and orthotopic neobladder (6%) three years after cystectomy, compared to those who received an ileal conduit (4% (p<0.001)), as for individuals with postoperative wound dehiscence (11% vs 4% without (p = 0.001)). At five years after cystectomy the proportion of patients operated with robotic assisted cystectomy subjected to MIH repair was higher (17%) compared to individuals

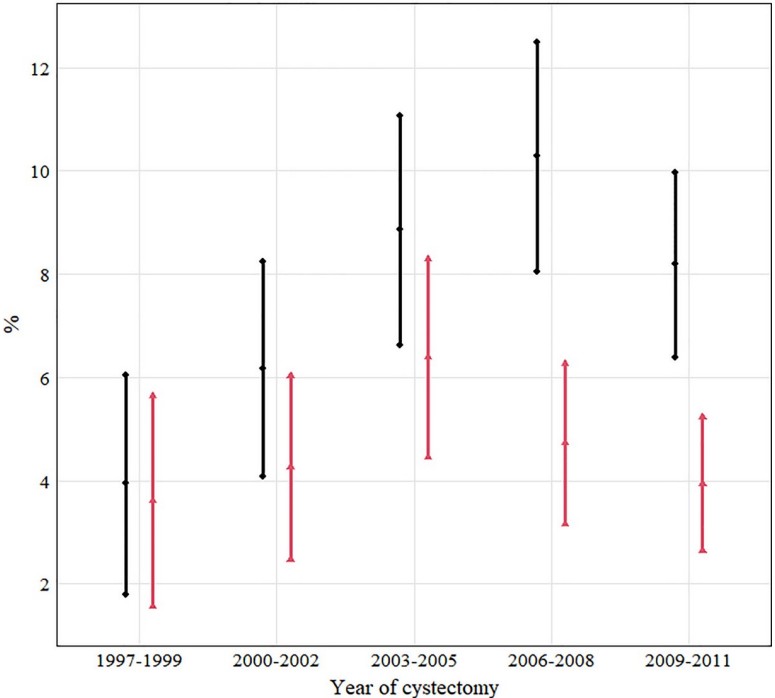

**Fig 2. Three year cumulative incidence with a 95% confidence interval.** Three year cumulative incidence midline incisional hernia diagnosis with 95% confidence intervals (black bars) and midline incisional hernia surgery (red bars), stratified by three-year time-periods. Patients with cystectomy during the last three-year strata 2012–2014 were excluded, as they did not have three years follow-up time.

operated with open cystectomy (6% (p = 0.005)). The other investigated potential determinants for MIH surgery displayed at similar proportions of such surgery irrespectively of each risk-factor (Table 1).

When pre- and perioperative risk-factors for MIH diagnosis was investigated in a Cox regression model, secondary cystectomy at disease progression after initially having an organ-sparing treatment strategy as primary treatment strategy (as opposed to primary cystectomy at bladder cancer diagnosis), continent cutaneous diversion, robotic assisted cystectomy, postoperative wound dehiscence, cystectomy during later years, and in hospitals with median cystectomy volume (PSMAV 10–25) were all independently associated with an increased risk in a multivariate analysis with almost similar univariate risks (Table 2).

Increased risk of MIH surgery was in a similar Cox-regression model associated with secondary cystectomy, continent cutaneous diversion, robotic assisted cystectomy, wound dehiscence and cystectomy during later years, both in univariate and multivariate analysis (Table 3). Wound dehiscence was associated with the most pronounced increased risk (HR 3.3 (2.0–5.4)), followed by a two-fold increased risk after continent cutaneous diversion and robotic assisted cystectomy (HR 2.1 (1.3–3.3) and HR 2.2 (1.2–3.9)), respectively.

In a subgroup analysis in patients operated with primary cystectomy where information on whether neoadjuvant chemotherapy was administered prior to surgery, the hazard ratio for MIH diagnosis after such preoperative treatment was 1.3 (0.8–2.1). Furthermore, among the

**Table 2. Cox regression univariate and multivariate analysis of known risk factors for subsequent incisional hernia diagnosis.**

| | Numbers | HR (univariate) | p-value | HR (multivariate) | p-value |
|---|---|---|---|---|---|
| Gender (Male vs Female) | 4291 | 1 | | 1 | |
| | 1355 | 1.03 (0.79–1.35) | 0.8 | 1.05 (0.79–1.39) | 0.7 |
| Age at cystectomy (Per unit) | 5646 | 0.99 (0.98–1.00) | 0.2 | 0.99 (0.98–1.01) | 0.4 |
| CCI | | | | | |
| 0 | 4006 | 1 | | 1 | |
| 1 | 761 | 1.21 (0.87–1.67) | 0.3 | 1.21 (0.87–1.70) | 0.3 |
| 2 | 556 | 1.09 (0.73–1.61) | 0.7 | 1.17 (0.78–1.74) | 0.4 |
| 3 | 145 | 0.63 (0.23–1.68) | 0.4 | 0.67 (0.25–1.80) | 0.4 |
| >3 | 117 | 0.90 (0.33–2.41) | 0.8 | 0.95 (0.35–2.57) | 0.9 |
| missing | 61 | | | | |
| Previous laparotomy (Yes vs No) | 56 | 0.42 (0.06–3.00) | 0.4 | 0.38 (0.05–2.77) | 0.3 |
| | 5590 | 1 | | 1 | |
| Previous inguinal hernia repair (Yes vs No) | 131 | 0.86 (0.48–1.53) | 0.6 | 0.88 (0.49–1.59) | 0.7 |
| | 5515 | 1 | | 1 | |
| Incisional hernia before cystectomy (Yes vs No) | 35 | 2.54 (0.82–7.93) | 0.1 | 2.77 (0.87–8.70) | 0.09 |
| | 5613 | 1 | | 1 | |
| Primary vs secondary cystectomy | 4108 | 1 | | 1 | |
| | 1538 | 1.40 (1.11–1.78) | 0.005 | 1.30 (1.02–1.67) | 0.04 |
| Urinary diversion | | | | | |
| Ileal conduit | 4347 | 1 | | 1 | |
| Orthtopic neobladder | 870 | 1.19 (0.89–1.59) | 0.3 | 1.14 (0.81–1.60) | 0.4 |
| Continent cutaneous pouch | 429 | 1.58 (1.12–2.24) | 0.01 | 1.92 (1.13–2.38) | <0.001 |
| Robot-assisted cystectomy (Yes vs No) | 346 | 1.41 (0.86–2.30) | 0.2 | 1.81 (1.04–3.15) | 0.04 |
| | 5300 | 1 | | 1 | |
| Wound dehiscence within 60 days after cystectomy (Yes vs No) | 181 | 2.70 (1.76–4.13) | <0.001 | 3.04 (1.96–4.71) | <0.001 |
| | 5465 | 1 | | 1 | |
| Year of cystectomy in three-year strata | | | | | |
| 1997–1999 | 468 | 1 | | 1 | |
| 2000–2002 | 719 | 1.61 (0.85–3.06) | 0.1 | 1.61 (0.84–3.06) | 0.2 |
| 2003–2005 | 908 | 2.38 (1.30–4.34) | 0.005 | 2.47 (1.34–4.55) | 0.004 |
| 2006–2008 | 1048 | 2.72 (1.51–4.90) | <0.001 | 3.08 (1.67–5.69) | <0.001 |
| 2009–2011 | 1234 | 2.20 (1.22–3.96) | 0.009 | 2.65 (1.43–4.92) | 0.002 |
| 2012–2014 | 1269 | 1.64 (0.86–3.10) | 0.1 | 1.71 (0.85–3.45) | 0.1 |
| PSMAV | | | | | |
| [0–10), | 1584 | 1 | | 1 | |
| [10–25), | 2206 | 1.44 (1.07–1.95) | 0.02 | 1.38 (1.01–1.87) | 0.04 |
| [25–86], | 1808 | 1.44 (1.05–1.97) | 0.02 | 1.15 (0.80–1.64) | 0.4 |
| missing | 48 | | | | |

PSMAV = Period-specific mean annual hospital volume, CCI = Charlson comorbidity index.

189 patients operated for postoperative ileus within 60 days of surgery the risk of MIH diagnosis was similar compared to those who did not (hazard ratio 1.1 (0.6–2.0)).

## Discussion

In this large population-based and nationwide cohort operated with radical cystectomy cumulative incidence midline incisional hernia (MIH) was eight percent and four percent were

**Table 3. Cox regression univariate and multivariate analysis of known risk factors for subsequent incisional hernia surgery.**

| | Numbers | HR (univariate) | p-value | HR (multivariate) | p-value |
|---|---|---|---|---|---|
| Gender (Male vs Female) | 4291 | 1 | | 1 | |
| | 1355 | 1.0 (0.72–1.38) | 1.0 | 1.01 (0.72–1.43) | 0.9 |
| Age at cystectomy (Per unit) | 5646 | 0.99 (0.97–1.00) | 0.1 | 0.99 (0.98–1.01) | 0.4 |
| CCI | | | | | |
| 0 | 4006 | 1 | | 1 | |
| 1 | 761 | 0.99 (0.65–1.51) | 1.0 | 1.10 (0.72–1.67) | 0.7 |
| 2 | 556 | 1.10 (0.69–1.75) | 0.7 | 1.21 (0.75–1.95) | 0.4 |
| 3 | 145 | 0.91 (0.34–2.44) | 0.8 | 1.01 (0.37–2.75) | 1.0 |
| >3 | 117 | 1.30 (0.48–3.51) | 0.6 | 1.42 (0.52–3.84) | 0.5 |
| missing | 61 | | | | |
| Previous | | | | | |
| laparotomy (Yes vs No) | 56 | 0.64 (0.09–4.54) | 0.7 | 0.72 (0.10–5.17) | 0.7 |
| | 5590 | 1 | | 1 | |
| Previous inguinal hernia repair (Yes vs No) | 131 | 0.82 (0.40–1.65) | 0.6 | 0.85 (0.41–1.74) | 0.7 |
| | 5515 | 1 | | 1 | |
| Primary vs secondary cystectomy | 4108 | 1 | 0.004 | 1 | 0.03 |
| | 1538 | 1.52 (1.14–2.01) | | 1.39 (1.04–1.86) | |
| Urinary diversion | | | | | |
| Ileal conduit | 4347 | 1 | | 1 | |
| Orthotopic neobladder | 870 | 1.34 (0.96–1.88) | 0.09 | 1.28 (0.86–1.90) | 0.2 |
| Continent cutaneous pouch | 429 | 1.74 (1.17–2.61) | 0.007 | 2.09 (1.34–3.26) | 0.001 |
| Robot-assisted cystectomy (Yes vs No) | 346 | 2.10 (1.26–3.51) | 0.004 | 2.16 (1.19–3.93) | 0.01 |
| | 5300 | 1 | | 1 | |
| Wound dehiscence within 60 days after cystectomy (Yes vs No) | 181 | 2.93 (1.81–4.76) | <0.001 | 3.28 (1.99–5.41) | <0.001 |
| | 5465 | 1 | | 1 | |
| Year of cystectomy in three-year strata | | | | | |
| 1997–1999 | 468 | 1 | | 1 | |
| 2000–2002 | 719 | 1.41 (0.72–2.79) | 0.3 | 1.41 (0.71–2.79) | 0.3 |
| 2003–2005 | 908 | 1.85 (0.97–3.51) | 0.06 | 1.92 (0.99–3.69) | 0.05 |
| 2006–2008 | 1048 | 1.68 (0.89–3.18) | 0.1 | 1.98 (1.01–3.88) | 0.05 |
| 2009–2011 | 1234 | 1.37 (0.73–2.60) | 0.3 | 1.69 (0.86–3.34) | 0.1 |
| 2012–2014 | 1269 | 1.98 (1.02–3.83) | 0.04 | 2.01 (0.96–4.23) | 0.06 |
| PSMAV | | | | | |
| [0–10), | 1584 | 1 | | 1 | |
| [10–25), | 2206 | 1.44 (1.01–2.06) | 0.05 | 1.31 (0.91–1.90) | 0.1 |
| [25–86], | 1808 | 1.44 (1.00–2.12) | 0.05 | 1.11 (0.71–1.73) | 0.6 |
| missing | 48 | | | | |

Previous incisional hernia was not adjusted for as no patient was operated after cystectomy with recurrent incisional hernia. PSMAV = Period-specific mean annual hospital volume, CCI = Charlson comorbidity index.

subjected to MIH repair at three years after cystectomy. As anticipated, MIHs were more frequently encountered in individuals who suffered from wound dehiscence after cystectomy, but also in patients receiving continent reconstructions compared to an ileal conduit. Patients operated with robotic assisted cystectomy were more frequently subjected to MIH repair at three and five years after cystectomy (8% and 17%, respectively), compared to patients operated with open cystectomy (four and six percent).

When adjusting for other pre- and perioperative risk-factors for MIH diagnosis, secondary cystectomy, continent cutaneous pouch diversion, robot-assisted cystectomy, wound dehiscence after cystectomy, cystectomy during the later parts of the study and cystectomy in hospitals with period-specific mean annual volume (PSMAV) in the middle tertile (10–25 annual cystectomies) were independently associated with an increased risk of MIH development. Information on surgeon volume was not available, and thus it was not possible to further disentangle the separate components (surgeon and hospital volume, respectively) of the association between PSMAV and risk of MIH diagnosis. Urinary diversion with an orthotopic neobladder has previously been reported to increase the risk of MIH [1], however orthotopic diversion were not associated with a decreased risk in the current study.

Our analysis shows that the cumulative incidence of MIH increases over the study period with no signs of a plateau in the curve (Fig 1). This implies that our estimates are difficult to compare to series reporting only crude proportions which do not consider censoring and a specific time horizon [1]. The continued risk over time illustrates the need for relevant time-to-event analysis of this complication after cystectomy, and that follow-up for at least three years is mandatory in any study evaluating MIH [7]. Furthermore, the reported risks of MIHs also depend on the methods of follow-up. The reported hernias in this study probably represent the great majority of clinically relevant MIH diagnosed, but when abdominal wall surgeons systematically reviewed postoperative abdominal computed tomographies in a prospective setting, MIHs were detected as frequently as in 55% of all patients [8]. Thus, the use of abdominal wall directed reviews of radiological examinations will not only increase the chance of detection, but also define a group of asymptomatic hernias. A thorough clinical examination has been suggested to be equally effective as radiological examinations for the detection of clinically relevant MIHs [9, 10].

The increased risk of MIH observed during the later studied three-year periods is difficult to explain. I.e. in the adjusted Cox model, it could not be explained by the increased proportions of patients with comorbidities. However, increased awareness of the complications over time and improved reporting may be reasons for such increased risk. Similarly, the finding that patients with MIH were younger than those without MIH in the present study is difficult to explain, but as age not were not associated with increased risk of MIH either as reported in other studies [11], probably selection mechanisms and unknown confounders contributed to this finding.

MIH as a frequent complication after continent cutaneous diversion has previously been reported, with such hernias diagnosed in eight percent of patients operated with continent cutaneous diversion ad modum Lundiana pouch in a recent single centre series [12]. In patients diverted with an orthotopic neobladder, difficulties to empty the substitute and use of excessive abdominal straining has also been associated with MIH development [3], and during the first part of the study orthotopic neobladders were constructed ad modum Goldwasser [13], also necessitating a longer midline incision as in Lundiana pouches, to mobilise the right colonic flexure compared to reconstructions using small intestine only. Furthermore, increased proportion of reoperations after a continent reconstruction related to long-term functional complications compared to an ileal conduit (29% vs 22%) [14], might also contribute and explain increased risk of MIH in these patients.

The increased risk of MIH diagnosis in patients operated in intermediate volume hospitals when adjusting for other risk-factors is probably explained by patient selection not captured in the available data. More thorough follow-up and detection and/or registration of MIH in these hospitals might contribute, especially as no increased risk of MIH surgery was noted (Table 3). The increased risk of MIH diagnosis and as well as MIH surgery associated with cystectomy performed as secondary treatment, compared to when cystectomy was performed at the

timepoint of bladder cancer diagnosis (primary cystectomy), is difficult to explain. For example, only 21/1538 patients treated with secondary cystectomy received radiotherapy prior to cystectomy, thus not a likely explanation for the increased risk of MIH. However, residual confounding and selection mechanisms in individuals whom postponing radical surgery in high-risk non-muscle invasive disease until later, could be tentative differences.

The counter-intuitive finding of a higher risk of MIH diagnosis and surgery after robotic assisted cystectomy was similar to after robot-assisted radical prostatectomy compared to open surgery [15]. The explanation for why shorter incisions applied in robotic assisted surgery were associated with increased risk of MIHs is not known, however explanations could be that "unconventional" incisions may have been used for specimen retrieval resulting in suboptimal fascial closure or earlier strenuous physical activities enabled by less wound pain after robotic assisted surgery provoking development of MIHs. Whether increased intraabdominal pressure per se during robotic assisted cystectomy is associated with the development of MIH is not known. As no information on whether intracorporeal reconstruction or extracorporeal reconstruction was applied during robotic assisted cystectomy, eventual different propensities for incisional hernia development between the two reconstruction methods could not be assessed in the current study. Furthermore, the robotic assisted cystectomies in the present series include the complete learning curve in all hospitals as opposed to a recent tertiary-care referral cohort without obvious learning curve within the series where robotic assisted cystectomy was not associated with an increased risk of MIH [16]. Thus, the possibility of longer operating times related to a steeper learning curve for robotic assisted cystectomy might also contribute to the higher risk of MIH diagnosis after robotic assisted cystectomy apart from the learning curve in itself and more careful follow-up of individuals operated with a new surgical technique.

A limitation of this study is the lack of information on the technique for wound closure. For example, a monofilament suture instead of multifilament reduces the risk of MIH [17], and recent European Hernia Society (EHS) guidelines recommends slowly absorbable monofilament suture in a single layer aponeurotic closure technique with small bites and a suture to wound length ratio at least 4:1 to decrease the risk of MIH [18]. Another limitation is the relatively low number of events entailing a low statistical precision concerning analyses of potential risk factors for MIH and MIH surgery, despite that the current study to our knowledge is the largest modern population-based series. Furthermore, one limitation is the lack of information on body mass index, smoking, corticosteroid medication, and extension of the incision above the umbilicus were not known and thus not possible to adjust for [19, 20]. The lack of information on intraoperative complications, blood loss at surgery and surgical duration and details on postoperative complications such as wound site infections and persistent postoperative anemia and hypoalbuminemia (both pre- and postoperatively) are also to be considered as study limitations. Likewise, there might be a possible underreporting of patients with MIHs into the patient registry. However, it is likely that the registered abdominal wall hernias constitute the vast majority of the advanced and symptomatic hernias and that the information on MIH hernia surgery from the patient registry has a high validity, as the procedure is associated with hospital reimbursement [20]. This assumption is also supported by similar proportions as in the current study were subjected to MIH repair in a population-based chart-review study (five percent of patients were operated for parastomal or MIH after cystectomy) [14] and a tertial referral center report (seven percent) [1]. Based on the continuously rising cumulative incidence curves for both MIH diagnosis and repair in Fig 1, a longer median follow-up than the current 2.3 years will likely add new events, which also can be considered a study limitation.

## Conclusions

The increased risk of MIH over time necessitates time-to-event analyses when studying this complication. Postoperative wound dehiscence was as anticipated a major risk-factor for MIH, emphasizing the importance of a technically optimal midline closure at primary surgery. The two hypothesis-generating findings that continent cutaneous diversion and robotic assisted cystectomy might be independent risk-factors for MIH, warrant prospective clinical studies.

## Author Contributions

**Conceptualization:** Fredrik Liedberg, Oskar Hagberg, Firas Aljabery, Truls Gårdmark, Staffan Jahnson, Tomas Jerlström, Agneta Montgomery, Amir Sherif, Viveka Ströck, Christel Häggström, Lars Holmberg.

**Data curation:** Fredrik Liedberg, Oskar Hagberg, Tomas Jerlström, Agneta Montgomery, Christel Häggström, Lars Holmberg.

**Formal analysis:** Fredrik Liedberg, Oskar Hagberg, Firas Aljabery, Truls Gårdmark, Staffan Jahnson, Tomas Jerlström, Agneta Montgomery, Amir Sherif, Viveka Ströck, Christel Häggström, Lars Holmberg.

**Funding acquisition:** Fredrik Liedberg, Lars Holmberg.

**Investigation:** Fredrik Liedberg, Oskar Hagberg, Firas Aljabery, Truls Gårdmark, Staffan Jahnson, Tomas Jerlström, Agneta Montgomery, Amir Sherif, Viveka Ströck, Christel Häggström, Lars Holmberg.

**Methodology:** Fredrik Liedberg, Oskar Hagberg, Firas Aljabery, Truls Gårdmark, Staffan Jahnson, Tomas Jerlström, Agneta Montgomery, Amir Sherif, Viveka Ströck, Christel Häggström, Lars Holmberg.

**Project administration:** Fredrik Liedberg, Oskar Hagberg, Firas Aljabery, Truls Gårdmark, Staffan Jahnson, Tomas Jerlström, Agneta Montgomery, Amir Sherif, Viveka Ströck, Christel Häggström, Lars Holmberg.

**Resources:** Fredrik Liedberg, Tomas Jerlström, Christel Häggström, Lars Holmberg.

**Supervision:** Fredrik Liedberg, Lars Holmberg.

**Validation:** Fredrik Liedberg, Oskar Hagberg, Truls Gårdmark, Christel Häggström, Lars Holmberg.

**Visualization:** Fredrik Liedberg, Oskar Hagberg, Christel Häggström, Lars Holmberg.

**Writing – original draft:** Fredrik Liedberg, Firas Aljabery, Truls Gårdmark, Staffan Jahnson, Tomas Jerlström, Agneta Montgomery, Amir Sherif, Viveka Ströck, Christel Häggström, Lars Holmberg.

**Writing – review & editing:** Fredrik Liedberg, Oskar Hagberg, Firas Aljabery, Truls Gårdmark, Staffan Jahnson, Tomas Jerlström, Agneta Montgomery, Amir Sherif, Viveka Ströck, Christel Häggström, Lars Holmberg.

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
