## [Decision Letter · Decision Letter 0]

8 Jan 2021

PONE-D-20-33796

Cumulative incidence of midline incisional hernia and its surgical treatment after radical cystectomy and urinary diversion for bladder cancer: A nation-wide population-based study

PLOS ONE

Dear Dr.Liedberg,

Thank you for submitting your manuscript to PLOS ONE. After careful consideration, we feel that it has merit but does not fully meet PLOS ONE’s publication criteria as it currently stands. Therefore, we invite you to submit a revised version of the manuscript that addresses the points raised during the review process.

Comments and questions;

1- Please comment and add information about postoperative ‘’ileus’’ incidence and correlation with MIH.

2-  There is some punctuation and printing mistakes in the text as in page 3 line  99 ‘’preforming’’!.  Please check and consider for editing and proofreading.

3- Please add comment about follow-up time that is 2.3 years. Is it sufficient or not? 

4- I would like to know that did ERAS protocol use in study group If yes, what was ERAS protocol a relation with MIH?

5- In discussion section, ‘’limitation’’ part must be in last paragraph of this section.

6-In discussion section, you have to discuss your results with literature; for example in seconda paragraph; there is no reference!

7- In this study authors reported that robotic surgery increased risk of MIH by two fold. Please ad a comment about intrabdominal pressure and its effects on radical cystectomy and also MIH.

8- In table 1a, please correct in order for ‘’Previous inguinal hernia repair; No , Yes ...results. 

9- The resolutions of Figure 1a and 1b are insufficient; I could not read the details of these figures.

We look forward to receiving your revised manuscript.

Kind regards,

Emre Bozkurt

Academic Editor

PLOS ONE

Additional Editor Comments:

Dear author,

Thank you for your well-designed manuscript. It needs some corrections before acceptance for publishing.

1- Please comment and add information about postoperative ‘’ileus’’ incidence and correlation with MIH.

2- There is some punctuation and printing mistakes in the text as in page 3 line 99 ‘’preforming’’!. Please check and consider for editing and proofreading.

3- Please add comment about follow-up time that is 2.3 years. Is it sufficient or not?

4- I would like to know that did ERAS protocol use in study group If yes, what was ERAS protocol a relation with MIH?

5- In discussion section, ‘’limitation’’ part must be in last paragraph of this section.

6-In discussion section, you have to discuss your results with literature; for example in seconda paragraph; there is no reference!

7- In this study authors reported that robotic surgery increased risk of MIH by two fold. Please ad a comment about intrabdominal pressure and its effects on radical cystectomy and also MIH.

8- In table 1a, please correct in order for ‘’Previous inguinal hernia repair; No , Yes ...results.

9- The resolutions of Figure 1a and 1b are insufficient; I could not read the details of these figures.

Journal Requirements:

2. Please include your tables as part of your main manuscript and remove the individual files. Please note that supplementary tables should be uploaded as separate "supporting information" files.

3. In your ethics statement in the Methods section and in the online submission form, please provide additional information about the data used in your retrospective study. Specifically, please ensure that you have discussed whether all data were fully anonymized before you accessed them and/or whether the IRB or ethics committee waived the requirement for informed consent. If patients provided informed written consent to have data from their medical records used in research, please include this information.

Reviewers' comments:

Reviewer's Responses to Questions

**Comments to the Author**

1. Is the manuscript technically sound, and do the data support the conclusions?

Reviewer #1: Yes

Reviewer #2: Yes

2. Has the statistical analysis been performed appropriately and rigorously? 

Reviewer #1: Yes

Reviewer #2: Yes

3. Have the authors made all data underlying the findings in their manuscript fully available?

Reviewer #1: Yes

Reviewer #2: Yes

4. Is the manuscript presented in an intelligible fashion and written in standard English?

Reviewer #1: Yes

Reviewer #2: Yes

5. Review Comments to the Author

Reviewer #1: Line 139: "-" sign is missing in the URL, please correct the URL as https://www.R-project.org/

Line 227-247: Please move the paragraph about limitations of the letter to the final part of discussion section

Reviewer #2: Dear authors,

This is a valuable orginal article about incidence of midline incisional hernia in surgery after radical cystectomy and urinary diversion for bladder cancer from Sweden. Thank you for your great effort but there are some concerns about the study.

Comments to authors;

1- Please comment and add information about postoperative ‘’ileus’’ incidence and correlation with MIH.

2- There is some punctuation and printing mistakes in the text as in page 3 line 99 ‘’preforming’’!. Please check and consider for editing and proofreading.

3- Please add comment about follow-up time that is 2.3 years. Is it sufficient or not?

4- I would like to know that did ERAS protocol use in study group If yes, what was ERAS protocol a relation with MIH?

5- In discussion section, ‘’limitation’’ part must be in last paragraph of this section.

6-In discussion section, you have to discuss your results with literature; for example in seconda paragraph; there is no reference!

7- In this study authors reported that robotic surgery increased risk of MIH by two fold. Please ad a comment about intrabdominal pressure and its effects on radical cystectomy and also MIH.

8- In table 1a, please correct in order for ‘’Previous inguinal hernia repair; No , Yes ...results.

9- The resolutions of Figure 1a and 1b are insufficient; I could not read the details of these figures.

Sincerely yours

6. PLOS authors have the option to publish the peer review history of their article (what does this mean?). If published, this will include your full peer review and any attached files.

Reviewer #1: No

Reviewer #2: No

---

## [Author Response · Author response to Decision Letter 0]

20 Jan 2021

Dear Editor, 

Thank You for valuable input, relevant questions and comments on the manuscript “Cumulative incidence of midline incisional hernia and its surgical treatment after radical cystectomy and urinary diversion for bladder cancer: A nation-wide population-based study

 PONE-D-20-33796”. We have now adapted the text accordingly, and hope that the manuscript again can be considered for publication in PLOS ONE. The answers to the comments are below and marked in red, as are the changes in the revised marked-up manuscript.

Regarding data availability, the data in in BladderBaSe is partly available in annual reports from the Swedish National Registry of Urinary Bladder Cancer (SNRUBC) and also are accessible online at https://statistik.incanet.se/urinblasecancer/. Collaborators can propose and apply for studies in the BladderBaSe using a standardised form. After approved application, theproject data administrators can upload study-specific files with selected variables

to a server for statistical analysis through remote access. Users of this system will

be charged for software licences, data administration and for preprocessing of

study files. For more information contact either the PI BladderBaSe (lars.holmberg@kcl.ac.uk) or the corresponding author (fredrik.liedberg@med.lu.se).

On behalf of all authors, 

kind regards!

Fredrik Liedberg

Additional Editor Comments:

Journal Requirements:

Answer: According to the style templates above the following corrections have been made:

Figure citations have now been exchanged from Figure 1a and Figure 1b to “Fig 1a” and “Fig 1b”, respectively. 

Reference citations has been changed from () to “[]” throughout the manuscript.

Figure titles have been changed to bold type: “Fig 1a. Cumulative incidence with 95% CI at selected time points.” “Fig 1b. Three-year cumulative incidence midline incisional hernia and midline incisional hernia surgery.”

Acknowledgement section has been changed to Funding section.

The adress to the corresponding author has been adopted as follows: *Corresponding author: fredrik.liedberg@med.lu.se (FL)

2. Please include your tables as part of your main manuscript and remove the individual files. Please note that supplementary tables should be uploaded as separate "supporting information" files.

Answer: Tables 1a, 1b and 1c is now formatted as part of the main manuscript and Table titles have been changed to bold type: “Table 1a. Number of patients with and without midline incisional hernia diagnosis and surgery.” “Table 1b. Cox regression univariate and multivariate analysis of known risk factors for subsequent incisional hernia diagnosis.” “Table 1c. Cox regression univariate and multivariate analysis of known risk factors for subsequent incisional hernia surgery.”

3. In your ethics statement in the Methods section and in the online submission form, please provide additional information about the data used in your retrospective study. Specifically, please ensure that you have discussed whether all data were fully anonymized before you accessed them and/or whether the IRB or ethics committee waived the requirement for informed consent. If patients provided informed written consent to have data from their medical records used in research, please include this information.

Answer: The current study was perfomed in BladderBaSe that is a research-database with all data fully anonymized. BladderBaSe was collated without written consent from the participating individuals. To further add this information, the following sentence has been modified to: “All data used were fully anonymized during the study that was approved by the Research Ethics Board of Uppsala University, Sweden (File no. 2015/277). “

4. We note that you have indicated that data from this study are available upon request. PLOS only allows data to be available upon request if there are legal or ethical restrictions on sharing data publicly. For information on unacceptable data access restrictions, please see http://journals.plos.org/plosone/s/data-availability#loc-unacceptable-data-access-restrictions. In your revised cover letter, please address the following prompts:

If there are ethical or legal restrictions on sharing a de-identified data set, please explain them in detail (e.g., data contain potentially identifying or sensitive patient information) and who has imposed them (e.g., an ethics committee). Please also provide contact information for a data access committee, ethics committee, or other institutional body to which data requests may be sent.

Answer: The following data sharing statement has now been added in the revised cover letter above: “The data in in BladderBaSe is partly available in annual reports from the Swedish National Registry of Urinary Bladder Cancer (SNRUBC) and also are accessible online at https://statistik.incanet.se/urinblasecancer/. Collaborators can propose and apply for studies in the BladderBaSe using a standardised form. After approved application, the

project data administrators can upload study-specific files with selected variables

to a server for statistical analysis through remote access. Users of this system will

be charged for software licences, data administration and for preprocessing of

study files. For more information contact either the PI BladderBaSe (lars.holmberg@kcl.ac.uk) or the corresponding author (fredrik.liedberg@med.lu.se).”

If there are no restrictions, please upload the minimal anonymized data set necessary to replicate your study findings as either Supporting Information files or to a stable, public repository and provide us with the relevant URLs, DOIs, or accession numbers. Please see http://www.bmj.com/content/340/bmj.c181.long for guidelines on how to de-identify and prepare clinical data for publication. For a list of acceptable repositories, please see http://journals.plos.org/plosone/s/data-availability#loc-recommended-repositories.

Reviewer #1: 

Line 139: "-" sign is missing in the URL, please correct the URL as https://www.R-project.org/

Answer: The URL has now been corrected to: https://www.r-project.org

Line 227-247: Please move the paragraph about limitations of the letter to the final part of discussion section

Answer: Same comment as by Reviewer #2: The limitation part has now been moved to the last paragraph of this section. The references have been adjusted, accordingly.

Reviewer #2: 

Dear authors,

This is a valuable orginal article about incidence of midline incisional hernia in surgery after radical cystectomy and urinary diversion for bladder cancer from Sweden. Thank you for your great effort but there are some concerns about the study.

Comments to authors;

1- Please comment and add information about postoperative ‘’ileus’’ incidence and correlation with MIH.

Answer: The number (proportion (%)) of individuals that were operated for postoperative ileus within 60 days of cystectomy that later were operated for incisional hernia was 11/379 (3%) compared to 178/5267 (3%) in patients without postoperative ileus surgery, thus no difference. The corresponding hazard ratio for MIH diagnosis during follow-up after ileus surgery within 60 days of cystectomy was 1.07 (0.6-2.0). To add and comment this information, we have now added the following sentence in the results section: “Furthermore, among the 189 patients operated for postoperative ileus within 60 days of surgery the risk of MIH diagnosis was similar compared to those who did not (hazard ratio 1.1 (0.6-2.0)).”

2- There is some punctuation and printing mistakes in the text as in page 3 line 99 ‘’preforming’’!. Please check and consider for editing and proofreading.

Answer: “Preforming” has now been corrected to “performing”. 

3- Please add comment about follow-up time that is 2.3 years. Is it sufficient or not?

Answer: To further explain the possibility of more events with longer follow-up, the following sentence has been added in the discussion section: “Based on the continuously rising cumulative incidence curves for both MIH diagnosis and repair in Fig 1a, a longer median follow-up than the current 2.3 years will likely add new events, which also can be considered a study limitation.“

4- I would like to know that did ERAS protocol use in study group If yes, what was ERAS protocol a relation with MIH?

Answer: Unfortunately, the successive adoption to ERAS during cystectomy care in Sweden with over twenty pre-, intra- and postoperative ingredients makes it impossible to elucidate which patient in the current study that received some or all measures associated with ERAS. To explain this lack of information, the following sentence has been added in the material and methods section: “Information about whether early recovery after surgery (ERAS) measures were applied before, during and after radical cystectomy was lacking.”

5- In discussion section, ‘’limitation’’ part must be in last paragraph of this section.

Answer: Same comment as by Reviewer #1: The limitation part has now been moved to the last paragraph of this section. The references have been adjusted, accordingly.

6-In discussion section, you have to discuss your results with literature; for example in seconda paragraph; there is no reference!

Answer: To further put our results into the context of available literature, the following wording has been added in the second paragraph in the discussion section: “Urinary diversion with an orthotopic neobladder has previously been reported to increase the risk of MIH [1], however orthotopic diversion were not associated with a decreased risk in the current study.” Furthermore, regarding the finding that younger age was associated with increased risk of MIH in the current study but not in other studies, the following sentence and new reference Sanders DL, Kingsnorth AN. The modern management of incisional hernias. BMJ. 2012 May 9; 344:e2843.) have been added in the third paragraph in the discussion section that also previously lacked a reference putting our findings into the scientific context:” Similarly, the finding that patients with MIH were younger than those without MIH in the present study is difficult to explain, but as age not were not associated with increased risk of MIH either as reported in other studies [11], probably selection mechanisms and unknown confounders contributed to this finding.”

7- In this study authors reported that robotic surgery increased risk of MIH by two fold. Please ad a comment about intrabdominal pressure and its effects on radical cystectomy and also MIH.

Answer: To our knowledge there are no studies available reporting association between laparoscopy, pneumoperitoneum and later incisional hernias. To further comment this fact, the following sentence has been added in the discussion section: “Whether increased intraabdominal pressure per se during robotic assisted cystectomy is associated with the development of MIH is not known.”

8- In table 1a, please correct in order for ‘’Previous inguinal hernia repair; No , Yes ...results.

Answer: This typo has now also been corrected.

9- The resolutions of Figure 1a and 1b are insufficient; I could not read the details of these figures.

Answer: Figure 1a and 1b have now been adjusted and with better resolution.

---

## [Editor Report · Decision Letter 1]

25 Jan 2021

Cumulative incidence of midline incisional hernia and its surgical treatment after radical cystectomy and urinary diversion for bladder cancer: A nation-wide population-based study

PONE-D-20-33796R1

Dear Dr. Liedberg,

We’re pleased to inform you that your manuscript has been judged scientifically suitable for publication and will be formally accepted for publication once it meets all outstanding technical requirements.

Kind regards,

Emre Bozkurt

Academic Editor

PLOS ONE

---

## [Editor Report · Acceptance letter]

26 Jan 2021

PONE-D-20-33796R1 

Cumulative incidence of midline incisional hernia and its surgical treatment after radical cystectomy and urinary diversion for bladder cancer: A nation-wide population-based study 

Dear Dr. Liedberg:

I'm pleased to inform you that your manuscript has been deemed suitable for publication in PLOS ONE. Congratulations! Your manuscript is now with our production department. 

Kind regards, 

on behalf of

Dr. Emre Bozkurt 

Academic Editor

PLOS ONE